# Effects of Different Film Types on Cotton Growth and Yield under Drip Irrigation

Zhanli Ma [1,2,3,4], Jian Liu [1,2,3,4], Yue Wen [1,2,3,4], Wenhao Li [1,2,3,4], Yan Zhu [1,2,3,4], Libing Song [1,2,3,4], Yunguang Li [1,2,3,4], Yonghui Liang [1,2,3,4] and Zhenhua Wang [1,2,3,4,*]

1   College of Water Conservancy and Architectural Engineering, Shihezi University, Shihezi 832000, China; 18892997981@163.com (Z.M.); liu_jian@shzu.edu.cn (J.L.); 17699534686@163.com (Y.W.); lwh8510012@163.com (W.L.); zhuyan2020@shzu.edu.cn (Y.Z.); songlibing@shzu.edu.cn (L.S.); 17690598519@163.com (Y.L.); liangyh202105@163.com (Y.L.)
2   Key Laboratory of Modern Water-Saving Irrigation of Xinjiang Production & Construction Group, Shihezi University, Shihezi 832000, China
3   Technology Innovation Center for Agricultural Water and Fertilizer Efficiency Equipment of Xinjiang Production & Construction Corps, Shihezi 832000, China
4   Key Laboratory of Northwest Oasis Water-Saving Agriculture, Ministry of Agriculture and Rural Affairs, Shihezi 832000, China
*   Correspondence: wzh2002027@163.com; Tel.: +86-132-0109-3132

**Abstract:** To address residual plastic film in fields in which mulched drip irrigation technology is applied, a sprayable degradable film (consisting of 1–5% clay, 5–20% humic acid, 0.5–5% crosslinker, and 0.5–5% auxiliary and water) can be used as a superior alternative to the plastic film applied in drip-irrigated fields. A field experiment was conducted in Xinjiang, Northwest China, to test the impacts of five different mulching treatments (SF1, sprayable degradable film applied at 1900 kg ha$^{-1}$; SF2, sprayable degradable film applied at 1900 kg ha$^{-1}$; SF3, sprayable degradable film applied at 2500 kg ha$^{-1}$; PF, plastic film; and NF, no film mulching) on cotton growth and development, yield, and water use efficiency. The results showed that, compared to the NF treatment, sprayable degradable film mulching (SF1, SF2, and SF3) positively impacted the soil hydrothermal environment, promoted root growth, significantly increased plant height and leaf area, and enhanced physiological characteristics, which, in turn, increased yield and water use efficiency by 11.79–15.00% and 21.88–30.21%, respectively. The maximum yield and water use efficiency were observed in the PF treatment, amounting to 5345 kg ha$^{-1}$ and 1.28 kg m$^{-3}$, respectively, and they had no significant differences from those in the SF3 treatment. In general, applying moderate amounts of sprayable degradable film at a rate of 2500 kg ha$^{-1}$ represents an effective agronomic strategy for managing residual film contamination while maintaining stable cotton yields.

**Keywords:** sprayable degradable film; plastic film; cotton yield; physiological characteristics; water use efficiency; drip irrigation; Xinjiang

## 1. Introduction

Water resources are a fundamental element of agricultural development. Water shortage seriously restricts agricultural development and threatens global food security [1,2]. In China, the distribution of water resources is uneven both spatially and temporally, resulting in unbalanced supply and demand [3,4]. Water shortages, resulting from scarce precipitation and intense evaporation, severely restrict agricultural development in the Xinjiang region [5–7]. However, this area's abundant solar and thermal resources give it great potential for agricultural development [8]. To ensure sustainable agricultural development and a secure food supply, conserving irrigation water and enhancing crop productivity are crucial [9].

Mulched drip irrigation technology alleviates water shortages, increasing soil temperature and moisture and enhancing water and fertilizer utilization efficiency, thereby leading

to higher crop yields and increased income. Due to its success, this technology has been widely adopted in Xinjiang [10]. With ongoing refinements, Xinjiang has become China's leading producer of high-quality cotton and the largest consumer of plastic film [11,12]. Plastic mulch has limited strength and poor mechanical properties, resulting in most of the corresponding film becoming debris after just one reproductive period [13]. Furthermore, film-recycling technology remains nascent, with a recycling rate below 60% [12]. Over the long term, the soil in Xinjiang's cotton fields accumulates residual film, averaging 260 kg ha$^{-1}$ in the cultivated layer, which can hinder soil permeability, restrict root access to water and nutrients [14], impede soil microbial activity [15], and ultimately reduce crop yield and quality [13,16]. While mulched drip irrigation technology has significantly boosted yield and income, it has also posed several challenges for sustainable agriculture development [16–19]. To promote agricultural production and environmental protection, addressing the issue of residual film is essential.

To mitigate the soil pollution caused by plastic film residues, researchers have begun using alternative mulching techniques to replace traditional plastic films [20]. Currently, there are two primary methods of reducing the accumulation of residual films. One method involves using degradable films, such as biodegradable [21], photodegradable [22], oxo-biodegradable, and sprayable degradable films [23]. The other method involves reducing the amount of plastic film applied, such as via uncovering the plastic film and cultivating without mulch [24]. Sprayable degradable films, composed of a newly developed polymeric organic compound, form a black soil film on the soil's surface when sprayed with water [23]. It has been demonstrated that sprayable degradable films can provide numerous benefits for soil health and crop yield. Sprayable degradable film mulching can significantly improve soil water conditions by reducing soil evaporation, particularly at higher polymer application rates like 10,000 kg ha$^{-1}$ [25]. Soil evaporation was reduced by more than 60% at a low sprayable degradable film application rate with the use of an additional viscosity modifier [26]. Additionally, sprayable degradable film controlled weeds nearly as effectively as a conventional plastic film at an application rate of 5000 kg ha$^{-1}$ [27]. Applying sprayable degradable film mulching can significantly increase crop yields and farmers' incomes [28–30].

Film material and width can affect cotton yield and its quality. Previous research has focused heavily on solid mulch [21,22]. However, very little research has been conducted on sprayable degradable films. This study aims to assess how changes in the growth and yield of cotton are affected by sprayable degradable films, specifically with respect to (i) the soil hydrothermal environment; (ii) cotton growth and photosynthetic characteristics; and (iii) cotton yield and water use efficiency.

## 2. Materials and Methods

### 2.1. Experimental Site

This experiment was conducted at the Key Laboratory of Modern Water-Saving Irrigation of Xinjiang Production and Construction Crops (85°59′ E, 44°19′ N) in April–October 2014 in Xinjiang, China, where there is a mean sea level of 412 m and an average ground slope of 0.6%. The study area is subject to a typical arid continental climate. The mean number of annual sunshine hours is 2865 h, and the accumulated temperature above 10 °C is 3463.5 °C d. Precipitation and annual air temperature were 212.8 mm and 22.3 °C, respectively, during the cotton-growing season (April–October) in 2014. The monthly precipitation and air temperature during cotton growth are shown in Figure S1.

### 2.2. Experimental Materials

The sprayable degradable film used in this study was developed by Lanzhou Institute of Chemical Physics, Chinese Academy of Sciences, and the weight percentages of its components are as follows: clay, 1–5%; humic acid, 5–20%; crosslinker, 0.5–5%; and water and auxiliary, 0.5–5%. This plastic film is produced by Xinjiang Tianye Company (Shihezi,

China). Its main component is polyethylene, and its thickness is $(0.008 \pm 0.0003)$ mm. The cotton variety used was Xinluzao 48, which is widely planted in Xinjiang.

### 2.3. Experimental Design

In the experiment, we utilized barrel planting, with plastic barrel dimensions of 0.52 m × 0.45 m × 0.35 m (height × top inner diameter × bottom inner diameter). The soil type in the barrel was medium loam, with an average bulk density of 1.37 kg m$^{-3}$ and a field water capacity of 0.18 g g$^{-1}$. This study involved five treatments (SF1, in which 1900 kg of sprayable degradable film was applied per ha; SF2, in which 2200 kg of sprayable degradable film was applied per ha; SF3, in which 2500 kg of sprayable degradable film was applied per ha; PF, plastic film; NF, no film mulching) applied during the cotton-growing seasons in 2014. Each treatment was replicated three times. Seeds were sown on 21 April 2014 using the "dry sowing and wet sowing" method. An equilateral triangle was applied 15–20 cm from the center of each barrel, with 2–3 cotton seeds placed at each vertex. Each bucket was equipped with two drippers situated 7.5 cm from the inner wall (Figure 1). The amount of sprayable degradable film used was calculated according to the top area of the test bucket, with the film being weighed using an electronic scale, put into three containers, diluted with water in a 2:1 ratio, and evenly sprayed on the soil surface in the bucket using a conventional agricultural sprayer. The amount of plastic film used was the same as that used in general cotton fields (8.7 kg ha$^{-1}$). Deep groundwater was used for irrigation during the growth period, with a mineralization level of 1.3 g L$^{-1}$. The total irrigation quota was 378 mm, distributed over 11 events, each delivering approximately 35 mm. All other field management practices were conducted according to local standards.

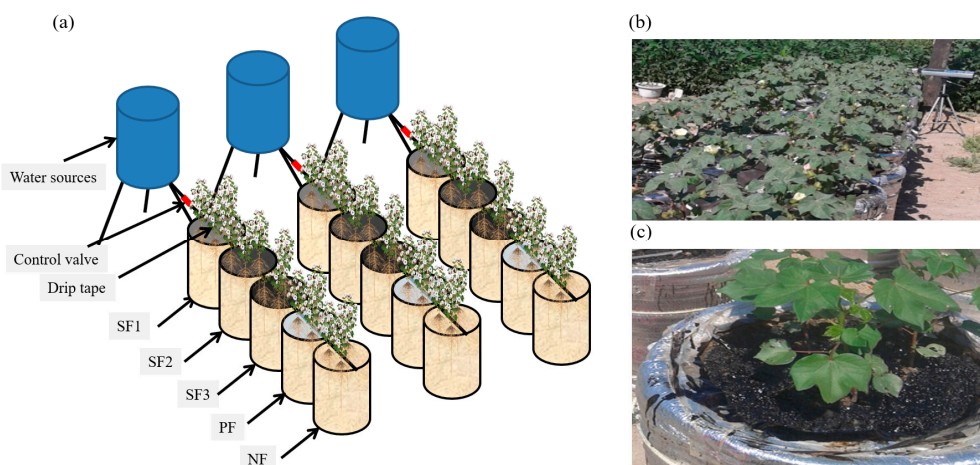

**Figure 1.** Schematic diagram of test layout (**a**,**b**) and application of sprayable degradable film (**c**).

### 2.4. Experimental Method

#### 2.4.1. Soil Water Content and Soil Temperature

Soil samples were taken from 0–10, 10–20, 20–30, and 30–40 cm layers at 3, 11, 14, 16, 19, 26, 33, and 40 DAS (days after sowing). Soil moisture was measured using the oven-drying method. Soil temperature was measured using curved alcohol thermometers (Hongxing Instruments Factory, Shihezi, China) at a depth of 0–10 cm at 3, 5, 12, 19, 28, and 35 DAS, respectively.

#### 2.4.2. Root Length Density and Root Weight Density

Root systems were removed using the lateral method after harvesting, with each 10 cm used as a sampling unit. All three barrels of each treatment were sampled as three replicates. Soil samples were soaked for 24 h and sieved through a 0.5 mm mesh, and all roots were extracted and photographed for total root length quantification via computerized image analysis. The samples were placed in an oven, dried at 105 °C for 0.5 h, turned at 75 °C

until a constant weight was achieved, and subsequently weighed on an electronic balance accurate to 0.1 mg.

### 2.4.3. Photosynthetic Properties

A CI-340 photosynthetic measurement system (CID, Portland, OR, USA) was used to measure the photosynthetic characteristics of leaves, including net photosynthetic rate (Pn, $\mu$mol $(m^2\ s)^{-1}$); intercellular $CO_2$ concentration (Ci, $\mu$mol $mol^{-1}$); stomatal conductance (Gs, mol $(m^2\ s)^{-1}$); and transpiration rate (Tr, mmol $(m^2\ s)^{-1}$). Measurements were consistently taken between 14:00 and 16:00 during each reproductive period under clear and cloudless conditions. At each measurement, the same leaf position and l age on the upper part of the main stem were selected, with the average value derived after three measurements.

### 2.4.4. Yield and Water Use

Water use efficiency (WUE) is the ratio of seed cotton yield to evapotranspiration (*ET*) at harvest, given in kg $m^3$. Evapotranspiration (*ET*) was measured according to the following water balance equation:

$$ET = 10 \cdot (W_1 - W_2)/(A \cdot \rho_w) + M + P_0 \tag{1}$$

where $W_1$ and $W_2$ are the bucket masses at the beginning and end of the period, given in g; *A* is the soil surface area in the bucket, given in $cm^2$, constituting 1590 $cm^2$ in this case; $\rho_w$ is the density of water, namely, 1.0 g $cm^3$; *M* is the irrigation water volume during the period, given in mm; and $P_0$ is the effective rainfall, given in mm.

### 2.5. Data Analysis

Variance analysis and path analysis were performed using SPSS 20.0 (SPSS Inc., Chicago, IL, USA). Significantly differences among various treatments were calculated according to the least significant difference (LSD) at the $p < 0.05$ level. Origin 2021b was used for drawing and function fitting. Excel 2010 was used for data collation.

### 3. Results

### 3.1. Soil Hydrothermal Environment

### 3.1.1. Soil Water Content

In the 0–40 cm soil layers, the soil water content was significantly higher in the mulched soils compared to those without mulch (Figure 2). Soil water content increased with soil depth. Compared with the NF treatment, the average soil water content in the 0–40 cm soil layers in the soils subjected to SF1, SF2, SF3, and PF increased by 16.16%, 18.76%, 20.30%, and 21.91%, respectively. Film mulching had a better performance in preventing soil moisture. However, the effectiveness of sprayable degradable film in conserving soil water content gradually waned over time. Soil water content in the soil subjected to SF1, SF2, and SF3 was increased by 17.18–21.29% at 3 DAS and 9.8–13.55% at 40 DAS, respectively.

### 3.1.2. Soil Temperature

The effects of different mulching technologies on the surface soil temperature from 3–35 DAS were significant (Table 1). Film mulching significantly increased soil temperature. Soil temperature was substantially higher, corresponding to an order ranging from PF, SF3, SF2, SF1, to NF. Compared to NF, the average soil temperature increased by 3.17%, 5.82%, 9.50%, and 14.50% in the SF1, SF2, SF3, and PF groups, respectively. Over time, the warming effect of spraying degradable film on soil gradually decreased. There was no significant difference in soil temperature between SF1 and NF treatments at 19–35 DAS.

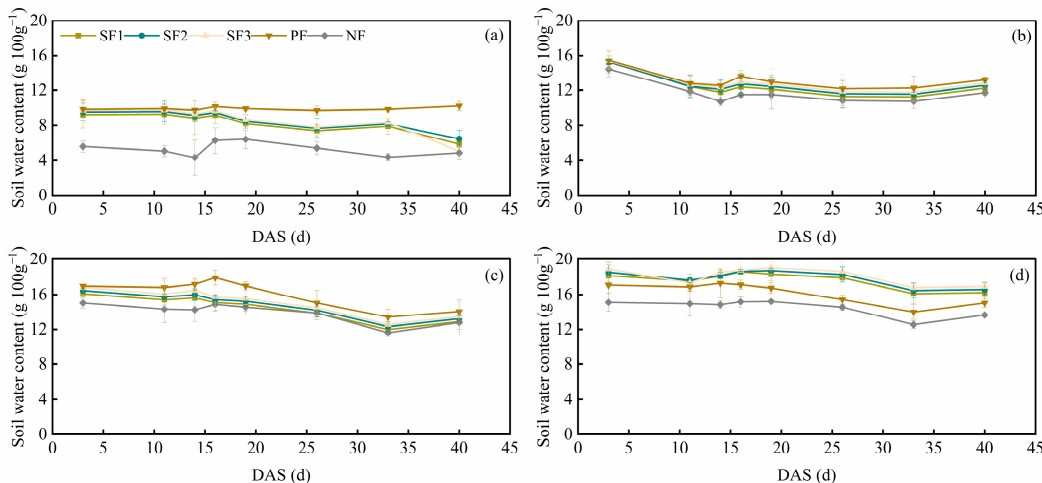

**Figure 2.** The soil water content in 0–40 cm soil depth from 3–40 DAS after applying different mulching treatments. SF1, SF2, SF3, PF, and NF represent 1900 kg of sprayable degradable film per ha, 2200 kg of sprayable degradable film per ha, 2500 kg of sprayable degradable film per ha, plastic film drip irrigation, and no film mulching, respectively. Note: Soil water content at 3, 11, 14, 16, 19, 26, 33, and 40 DAS in 0–10 (**a**), 10–20 (**b**), 20–30 (**c**), and 30–40 cm (**d**) soil layers under different mulching treatments, respectively. This applies below as well.

**Table 1.** Soil temperature in 0–10 cm soil layer under different treatments. Letters represent the significant differences at the $p < 0.05$ level determined using LSD.

| Treatments | Day after Sowing (d) | | | | | |
|---|---|---|---|---|---|---|
| | 3 | 5 | 12 | 19 | 28 | 35 |
| SF1 | 34.1 ± 0.3 c | 29.2 ± 0.4 c | 30.3 ± 0.2 d | 29.7 ± 0.6 bc | 28.3 ± 0.3 d | 21.2 ± 0.4 d |
| SF2 | 34.7 ± 0.4 c | 30.1 ± 0.5 b | 31.2 ± 0.4 c | 30.0 ± 0.4 bc | 29.2 ± 0.5 c | 22.0 ± 0.4 c |
| SF3 | 35.5 ± 0.5 b | 30.3 ± 0.2 b | 33.5 ± 0.4 b | 30.5 ± 0.5 b | 30.5 ± 0.4 b | 23.1 ± 0.4 b |
| PF | 36.6 ± 0.4 a | 31.5 ± 0.4 a | 35.5 ± 0.4 a | 31.5 ± 0.3 a | 31.9 ± 0.3 a | 24.8 ± 0.4 a |
| NF | 32.2 ± 0.4 d | 28.4 ± 0.3 d | 28.7 ± 0.5 e | 29.5 ± 0.5 c | 27.9 ± 0.6 d | 20.8 ± 0.2 d |

### *3.2. Cotton Growth*

#### 3.2.1. Plant Height

Plant height increased rapidly from 46 to 75 DAS, with an average growth rate of 9.37 mm d$^{-1}$, and then slowed after 75 DAS due to tip pruning (Figure 3). Different amounts of the sprayable degradable film significantly affected plant height ($p < 0.05$). Compared to NF, the plant height in the groups subjected to SF1, SF2, SF3, and PE increased by 2.08%, 5.28%, 10.28%, and 19.41% 75 DAS. Maximum plant heights for SF1, SF2, SF3, PF, and NF were recorded at 107 DAS, measuring 41.82, 43.67, 45.33, 46.67, and 40.50 cm, respectively. Plant height in PE and SF3 significantly differed from NF 46–127 DAS. The plant height reduction rate for the sprayable degradable film was less than that for NF and higher than that for PE 107–124 DAS.

#### 3.2.2. Leaf Area

Cotton leaf area gradually increased until 90 DAS and then steadily decreased (Figure 4). Different mulching treatments significantly enhanced the cotton leaf area, especially after 75 DAS ($p < 0.05$). Cotton leaf area was the highest after PF was applied, followed by SF3, SF2, SF1, and NF. The maximum leaf area corresponding to SF1, SF2, SF3, PF, and NF was recorded at 90 DAS, with values of 909.3, 1009.4, 1104.3, 1156.6, and 827.5 cm$^2$, respectively. Compared to NF, the cotton leaf area for SF1, SF2, SF3, and PE was increased by 9.23%, 21.27%, 32.66%, and 38.95% at 90 DAS. Therefore, sprayable degradable film mulching significantly increased leaf area, but the plastic film performed better.

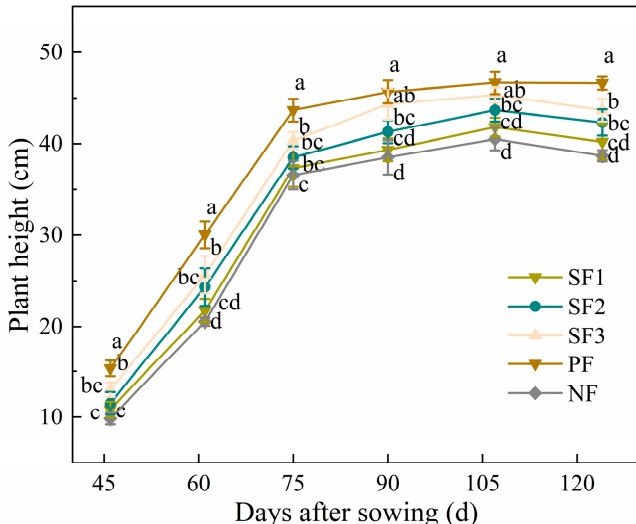

**Figure 3.** Plant height under different mulching treatments. The symbol is the mean of three repetitions; the vertical bar is the standard error. Letters represent significant differences at the $p < 0.05$ level determined using LSD.

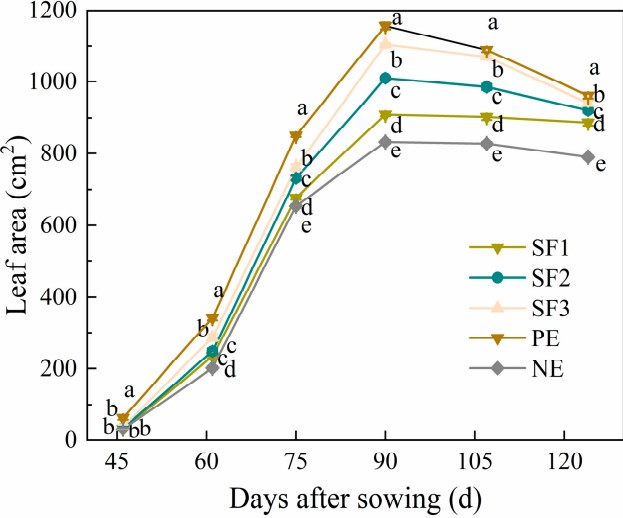

**Figure 4.** Leaf area under different mulching treatments. The symbol is the mean of three repetitions; the vertical bar is the standard error. Letters represent the significant differences at the $p < 0.05$ level determined using LSD.

### 3.2.3. Root Length Density and Root Weight Density

The sprayable degradable film influenced the root distribution of drip-irrigated cotton (Figure 5). Both root length density (RLD) and root weight density (RWD) increased with increasing amounts of sprayable degradable film. Compared to CK, the RLD corresponding to SF1, SF2, and SF3 increased by 32.98%, 57.18%, and 88.43%, and the RWD increased by 14.73%, 35.98%, and 68.17% for SF1, SF2, and SF3, respectively, within the 0–50 cm soil layer. Thus, sprayable degradable film mulching promoted the downward growth of cotton roots. In the 30–50 cm soil layer, the RWD of the sprayable degradable film mulching treatments showed an average increase of 39.63%. The RLD and RWD induced by SF3 were closer to those of PE, being 7.88% and 16.12% lower, respectively.

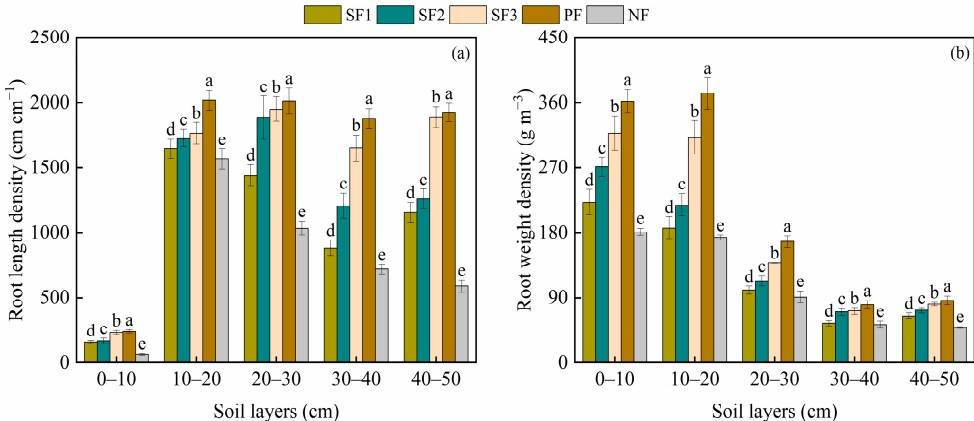

**Figure 5.** Root length density and root weight density of cotton under different mulching treatments at the boll-opening stage. (**a**) Root length density and (**b**) root weight density. The symbol is the mean of three repetitions; the vertical bar is the standard error. Letters represent the significant differences at the *p* < 0.05 level determined using LSD.

### *3.3. Photosynthetic Properties*

The average net photosynthetic rate (Pn), stomatal conductance (Gs), intercellular $CO_2$ concentration (Ci), and transpiration rate (Tr) were significantly influenced by the different mulching technologies across four stages (Figure 6). The sprayable degradable film improved the photosynthetic properties of cotton leaves. Across all stages, these four indicators followed the same ranking: PE > SF3 > SF2 > SF1 > NF.

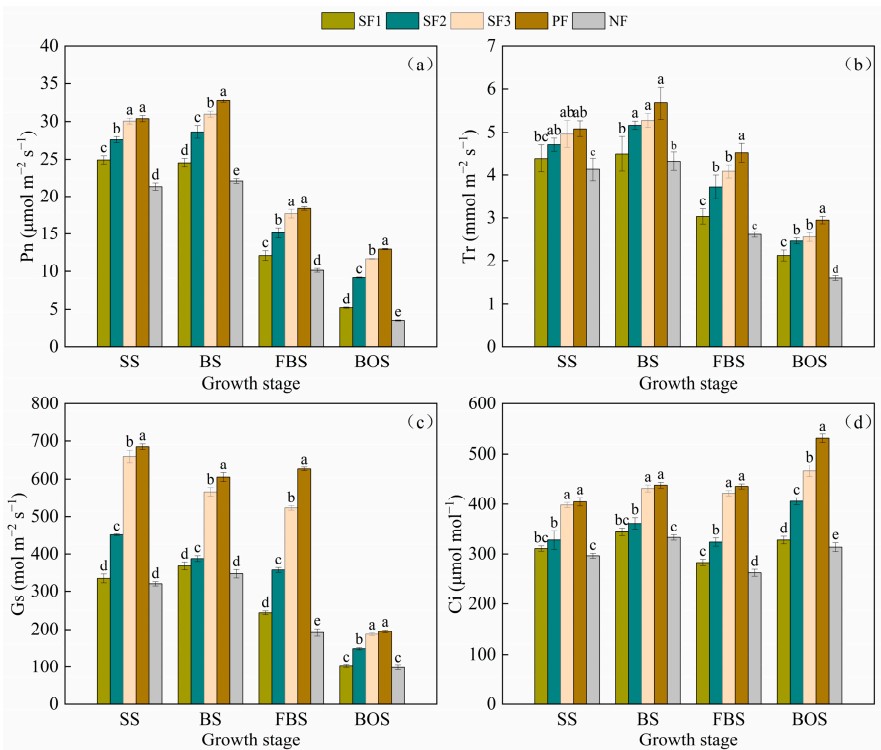

**Figure 6.** Photosynthetic properties of cotton under different mulching treatments at different stages. (**a–d**) Pn, Tr, Gs, and Ci represent the photosynthetic rate, transpiration rate, stomatal conductance, and intercellular $CO_2$ concentration. SS, BS, FBS, and BOS represent the seedling stage, budding stage, flower-bolling stage, and boll-opening stage, respectively. The symbol is the mean of three repetitions; the vertical bar is the standard error. Letters represent significant differences at the *p* < 0.05 level determined using LSD.

Overall, the Pn was slightly lower at the seedling stage than at the budding stage and gradually decreased after the budding stage (Figure 6a). Pn increased with an increasing amount of sprayable degradable film. At all stages, the Pn of cotton for SF1 and SF2 was significantly higher than that of NF and lower than that of PE. The Pn of cotton for SF3 was closer to that for PF. For example, the Pn values for SF1, SF2, SF3, and PF were higher than those for NF by 19.57%, 49.62%, 74.46%, and 81.78% at the flower-bolling stage.

Tr increased first, reaching its peak at the budding stage, and then decreased (Figure 6b). Only SF1 and NF showed significant differences from PF at the budding stage. Differences among treatments gradually increased. Tr showed a trend of increasing the amount of sprayable degradable film. The sprayable degradable film increased the Tr of cotton, though its enhancing effect diminished as the season progressed. The sprayable degradable film treatment showed significant differences compared to the PE treatment after the budding stage. Compared to PF, the Tr values corresponding to SF1, SF2, SF3, and NF were reduced by 26.02%, 9.75%, 7.46%, and 30.97%, respectively.

The effects of various application amounts of sprayable degradable film on the Gs of cotton throughout the reproductive period were significant (Figure 6c). The average Gs of cotton with respect to PF was the highest among the treatments, reaching 585.18 mol m$^{-2}$ s$^{-1}$. Compared to NF, the Gs of cotton in the SF1, SF2, SF3, and PF treatments increased significantly by 26.90%, 85.88%, 172.58%, and 226.73% at the flower-bolling stage, respectively. There was no significant difference between SF3 and NF at the boll-opening stage.

The average Ci of each treatment peaked at the boll-opening stage and was lowest at the boll stage (Figure 6d). The Ci increased with the increasing amounts of sprayable film applied; the Ci for SF3 was not significantly different from that for PE from the seedling stage to the budding stage.

### 3.4. Seed Cotton Yield and Water Use

Different mulching treatments significantly influenced seed cotton yield ($p < 0.05$) (Table 2). The highest and lowest seed cotton yields among the treatments were induced by PF and NF, with 5345 and 4648 kg ha$^{-1}$, respectively. Seed cotton yields increased with an increase in the amount of sprayable degradable film applied. Compared to NF, the seed cotton yields induced by SF1, SF2, and SF3 significantly increased by 11.79%, 14.39%, and 15.00% ($p < 0.05$), respectively. SF3 resulted in a lower cotton yield than PF, with no significant difference ($p > 0.05$).

**Table 2.** Effects of different mulching treatments on yield, ET, and WUE of cotton. Letters represent significant differences at the $p < 0.05$ level determined using LSD.

| Treatments | Yield (kg ha$^{-1}$) | ET (mm) | WUE (kg m$^{-3}$) |
|---|---|---|---|
| SF1 | 4980 c | 425.64 bc | 1.17 c |
| SF2 | 5196 b | 436.64 b | 1.19 bc |
| SF3 | 5317 a | 425.36 bc | 1.25 ab |
| PF | 5345 a | 417.58 c | 1.28 a |
| NF | 4648 d | 484.17 a | 0.96 d |

ET during the whole growing season was affected significantly by the various film-based mulching treatments (Table 2). The highest ET was recorded in relation to NF, and the lowest was in relation to PF. Different mulching treatments had significant effects on the WUE of cotton ($p < 0.05$) (Table 2). The WUE of sprayable degradable film mulching treatments increased with an increase in the amount applied, and all the values were significantly higher than those for NF. The highest and lowest WUE values among the treatments corresponded to PF and NF, with 1.28 and 0.96 kg m$^{-3}$, respectively. The WUE for SF3 was very close to that for the PF treatment; the difference was only 2.34%, and there were no significant differences between the values ($p > 0.05$).

### 3.5. The Relationship between Seed Cotton Yield and Growth Indicators

The growth status of the cotton root system directly impacts crop growth and yield formation. Root weight density demonstrated a significant linear correlation with leaf area, plant height, and seed cotton yield, with Pearson correlation coefficients of 0.979, 0.848, and 0.879, respectively (Table 3).

**Table 3.** Results of linear fitting of leaf area, plant height, and yield with root weight density.

| Indicators | Linear Fitting | Pearson Correlation | $R^2$ |
|---|---|---|---|
| Leaf area ($X_3$) | $y = 3.07X_3 + 521.88$ | 0.979 | 0.98 |
| Plant height ($X_4$) | $y = 0.070X_4 + 30.90$ | 0.848 | 0.72 |
| Yield ($X_5$) | $y = 6.13X_5 + 4138.47$ | 0.879 | 0.77 |

This path analysis reveals the complex relationships between plant characteristics and photosynthetic rate (Table 4). Leaf area has the highest simple correlation coefficient (1.913) and direct path coefficient (1.060), indicating a significant positive impact on photosynthetic rate. Similarly, root length density has a simple correlation coefficient of 0.974, a direct path coefficient of 0.190, and a total indirect path coefficient of 0.784, suggesting it primarily exerts a positive influence on photosynthetic rate through leaf area indirectly. In contrast, root weight density ($X_2$) has a negative direct path coefficient ($-0.279$) but a total indirect path coefficient of 1.243. Plant height ($X_4$) has a relatively minor impact on photosynthetic rate, with a total indirect effect of $-0.067$. Seed cotton yield is closely linked to photosynthetic rate, and there was a significant positive correlation between seed cotton yield and photosynthetic rate (Figure 7).

**Table 4.** Path analysis of the relationship between different indexes and photosynthetic rate.

| Independent Variable | Simple Correlation Coefficient | Direct Path Coefficient | Indirect Path Coefficient | | | | |
|---|---|---|---|---|---|---|---|
| | | | $X_1$ | $X_2$ | $X_3$ | $X_4$ | Total |
| Root length density ($X_1$) | 0.974 | 0.190 | — | $-0.278$ | 1.053 | 0.007 | 0.784 |
| Root weight density ($X_2$) | 0.964 | $-0.279$ | 0.189 | — | 1.047 | 0.007 | 1.243 |
| Leaf area ($X_3$) | 1.913 | 1.060 | 0.189 | $-0.275$ | — | 0.939 | 0.853 |
| Plant height ($X_4$) | $-0.058$ | 0.009 | 0.161 | $-0.236$ | 0.008 | — | $-0.067$ |

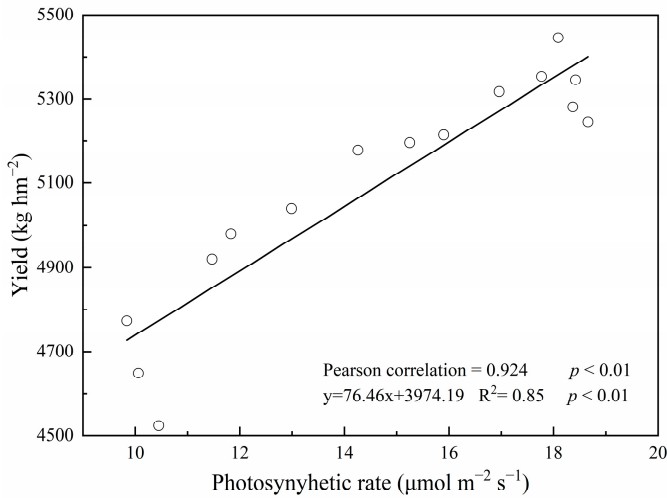

**Figure 7.** Relationship between yield and photosynthetic rate. Photosynthetic rate corresponds to the data for the flowering and boll stages, with five treatments and three replicates for each treatment.

## 4. Discussion

Plastic films are a popular choice for agricultural mulch due to their ability to enhance the soil environment in fields. Their benefits have established them as the preferred option for farmers seeking to improve crop yields and soil health [31]. However, growing concerns about the accumulation of residual film from plastic mulch, which poses a serious threat to sustainable agricultural development, have emerged [13,17,18]. A sprayable degradable film can degrade to $CO_2$, $H_2O$, and other non-toxic compounds under the action of microorganisms, avoiding the pollution of the soil [32]. Therefore, sprayable degradable films represent an effective alternative to plastic for soil ecology conservation.

High-density and evenly distributed root systems indicate well-developed roots. In particular, a high root mass, high activity, and numerous roots in the subsoil are the basis for achieving high cotton yields. Additionally, the soil environment plays a significant role in shaping the growth and development of the root system, ultimately impacting cotton yield [33–36]. Sprayable degradable films positively impact the soil environment by forming a black curing film on the soil surface after application, which can increase soil temperature and conserve moisture [27,29,37]. Adequate soil moisture and heat conditions improve root systems' growth and development [38,39]. Our research indicates that mulching improves the hydrothermal environment of soil, promoting root growth and laying the foundation for above-ground cotton plant growth and yield formation (Table 2). The sprayable degradable film mulching of cotton fields facilitated the downward development of cotton roots, increasing the percentage of lower roots, enhancing cotton's ability to absorb water and nutrients in the middle and late stages and preventing early cotton decline. However, the promoting effects of the sprayable degradable film were weaker than those of plastic mulch, as observed in previous studies [28,30].

Numerous studies have shown that root development is closely related to crop growth and yield [40–42]. This was also the case in our study, where there was a significant linear relationship between plant height, stem thickness, and yield and root weight density (Table 3). A greater amount of root biomass and length can stimulate plants to absorb a larger volume of water and nutrients from the soil. This, in turn, can enhance the growth of above-ground plants and ultimately result in higher crop yields [43–46]. Sprayable degradable film mulching promoted the growth of the cotton root system, which also provided a good foundation for above-ground growth and attaining high yields. Our study indicates that the use of sprayable degradable film under drip irrigation can increase the plant height and leaf area of cotton in the early stage and moderately suppress the growth rate of leaf area in the middle and late stages, thus laying a solid foundation for achieving higher seed cotton yields, similar to the results of previous studies [30,47].

A higher photosynthetic rate is significantly related to an increase in crop yield [48]. Path analysis has been widely adopted to analyze the interaction between a dependent variable and multiple independent variables via a linear regression model [49]. This study explored the relationship between photosynthetic rate and factors such as root length density, root weight density, and plant height. Our findings indicate that leaf area had the strongest direct impact on the photosynthetic rate, followed by root length density, plant height, and root weight density. (Table 4). The effects of leaf area on photosynthetic rate were both direct and indirect (Table 4). Improved management practices are considered to be an effective way in which to increase leaf photosynthetic activity by regulating crop growth conditions [50]. The mulching technique is known to promote stomatal opening and crop transpiration, which, in turn, improves photosynthesis. It is important to note that leaves have a direct influence on stomatal opening and crop transpiration [51]. The amount of the sprayable degradable film applied significantly affected the photosynthetic characteristics of drip-irrigated cotton leaves. The Pn, Tr, Gs, and Ci of the leaves increased with an increase in the amount of the sprayable degradable film applied, especially in the early stages of cotton growth. Sprayable degradable films begin to degrade towards the end of cotton growth, causing the mulching effect to gradually diminish. As a result, water use

efficiency and cotton yield are impacted as the films continuously alter the physiological characteristics of the cotton plants on which they are applied.

This study found that as the amount of sprayable degradable film applied increased, cotton root length density, root weight density, plant height, leaf area, photosynthetic properties, and yield also increased/improved. Additionally, the gap between these measurements and those of plastic mulch decreased gradually. Treatments in which low amounts were utilized resulted in poor film formation quality, low water retention, and insulation. These factors were not conducive to root growth, ultimately affecting the growth and yield of cotton. Previous studies have shown a decreasing trend in cotton yield, primarily caused by the excessive use of sprayable degradable films and poor film formation quality. These factors negatively impacted cotton emergence, resulting in a reduction in production efficiency [47,52]. The difference in this study was that the maximum amount of sprayable degradable film applied (2500 kg ha$^{-1}$) did not reach the upper limit of the application amount. One study showed that the optimal amount of sprayable degradable film was 112.5 kg ha$^{-1}$ [47], which is much lower than the 2500 km ha$^{-1}$ used in this study, mainly because of the low concentration of sprayable degradable film used in this study. Further research should consider economic and ecological benefits to comprehensively evaluate the optimal amount of sprayable degradable films to apply in drip-irrigated cotton fields.

## 5. Conclusions

In this study, we conducted a field experiment to investigate the impact of varying amounts of sprayable degradable film on the soil hydrothermal environment, cotton growth, physiological characteristics, yield, and water use efficiency under drip irrigation in a typical dryland area in Xinjiang, China. This study found that the application of sprayable degradable film had a positive impact on various aspects of cotton growth. Specifically, it improved the hydrothermal environment of the soil, leading to enhanced root growth and increased height and leaf area of cotton plants. Additionally, the films improved cotton's photosynthetic characteristics, resulting in higher yields and improved water use efficiency. The impact of sprayable degradable films on the physiological characteristics of cotton was most significant during the early growth stages. Furthermore, the positive effects of the sprayable film increased as the amount applied increased. The treatment with the highest amount of sprayable film had a yield and water use efficiency that were comparable to those of the plastic mulch treatment, with no significant differences observed. When applying 2500 kg ha$^{-1}$ of a sprayable degradable film, the combination of a sprayable degradable film and drip irrigation can have water-saving and yield-increasing effects comparable to those of drip irrigation under plastic mulch. According to the experimental results, no less than 2500 kg ha$^{-1}$ of sprayable degraded should be applied in the Xinjiang cotton area. The findings of this study offer valuable insights into the possibility of substituting plastic mulch with sprayable degradable film. Additionally, this research will provide theoretical guidance for promoting sustainable cotton production in the Xinjiang region. Sprayable degradable films' biodegradability and non-polluting properties make it an incredibly promising technology for use in similar dryland regions.

**Supplementary Materials:** The following supporting information can be downloaded at: https://www.mdpi.com/article/10.3390/su16104173/s1. Figure S1. Monthly temperature and precipitation during cotton-growing periods.

**Author Contributions:** Conceptualization, Y.L. (Yonghui Liang), Y.L. (Yunguang Li) and W.L.; Methodology, Z.W.; Software, Y.Z.; Resources, L.S.; Data curation, J.L.; Writing—original draft preparation, Z.M.; Writing—review and editing, Z.M., J.L., Z.W. and Y.W.; Project administration, Z.W. and J.L. All authors have read and agreed to the published version of the manuscript.

**Funding:** This work was funded by the National Key R&D Program of China (No. 2021YFD1900802-2, Jinzhu Zhang), the National Natural Science Foundation of China (No. 52169012, Zhenhua Wang; 52169011, Libing Song), the Key Areas of Science and Technology Research Program (No. 2022AB011-01, Jinzhu Zhang), the Scientific Research Foundation for High-level Talents, Shihezi University (No.

2022ZK010), the Innovation and Development Project, Shihezi University (No. CXFZ202106, Jian Liu), and the "Tianchi talent" Introduction Program-Young Doctor-Liu Jian.

**Data Availability Statement:** The data presented in this study are available on request from the corresponding author.

**Conflicts of Interest:** The authors declare no conflicts of interest.

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
