# Peer review of "Effects of Different Film Types on Cotton Growth and Yield under Drip Irrigation"

_sustainability, doi:10.3390/su16104173_

Round 1

Reviewer 1 Report

Comments and Suggestions for Authors

Thank you for this research, which is important and well conducted. Your conclusions are satisfactory although comment on (i) the practical ability of applying >2 t/ha of the sprayable degradable mulch onto broadacre cotton production and (ii) its biodegradability would be valuable. These will be considerations in its uptake.

Comments on the Quality of English Language

Minor editing of the English expression would help the paper's readability.

Reviewer 2 Report

Comments and Suggestions for Authors

Dear authors,

This paper deals with the effects of sprayable degradable film mulching on cotton growth and yield under drip irrigation. There is a coherence between the hypothesis of the work, the methodology and the results presented. The abstract is adequate and the conclusions are in agreement with the results presented. However, there are different issues that should be addressed before being considered for publication.

Abstract section. Composition of the sprayable biodegradable film, must be included.

Introduction section.

It is informative but previous work using this material should be mentioned and results disscussed.

Methodology section. 

Line 77... consider to change the word ... subsection.. for a more accurate term.

Figure 1. is not necessary in this section (maybe Supplementary).

Figure 2. Must include a photography of the product applied to the ground... to assure it form a suitable structure you called film mulch. Due to the film mulch chemical composition that you state here...there is no polymeric material that could form films. Thus it is mandatory to show the apparience of the mulch film. 

Results.

It is necessary to include in tables or figures names the statistical analysis that was performed on each case. 

It is suggested that Table 3 and 4 were placed in Results section and not in Discussion section.

Figure 6 can be presented in a normal X-Y graphic... with Y axes at the right position. Statistical analysis is required.

Reviewer 3 Report

Comments and Suggestions for Authors

This study explored the effects of different film applications on soil properties, cotton growth and water use. The background, research design and results were properly stated. But the soil moisture and temperature data were only measured up to 35 days after sowing, while the cotton growth season lasts several months. Especially under sprayable degradable film treatments, the film coverage changed over time, which may significantly affect the soil environment.

Abstract

Lines 15-16: The sprayable and degradable film is a superior alternative to plastic film applied in drip irrigated fields.

Line 17: impacts of five different mulching treatments (SF1, SF2, …) on…

Lines 24-25: this result is contradictory to the previous result which indicated that sprayable degradable film treatments improved the cotton yield and water use efficiency.

Introduction

Lines 58-59: delete some researchers or change into researches.

Materials and Methods

Line 77: subsection change into experimental site.

Line 82: annual temperature would be better.

Lines 82-83: to address the months of the year for cotton growing season.

Figure 1: I’m confused, is the cotton planted in the pots? If it is, how many pots did each treatment have?

Line 119: what’s the frequency of soil sample collection?

Line 120: oven drying?

Results

Line 155: increases change into increased.

Figure 3: why the soil water content and temperature were measured only up to 35 days after sowing, but the cotton growing season lasts several months.

Lines 233-234: not clear, rewrite this sentence.

Line 246: what does rose mean?

Discussion

Lines 285-286: reorganize this sentence.

Line 315: Path analysis was not described in data analysis section.

Comments on the Quality of English Language

Language needs moderate editing.

Reviewer 4 Report

Comments and Suggestions for Authors

I got opportunity to review the research article submitted by Ma et al., 2024 entitled: Effects of sprayable degradable film mulching on cotton growth and yield under drip irrigation for possible publication in Sustainability. The performed work comes under the scope of journal and special issue. In addition, after carefully reviewing and reading the manuscript, I came to conclude that manuscript is not technically sound to be considered for possible publication because only one cropping seasons data is not sufficient to justify the test results. As there are several negative effects of the plastic application on soil and plant for long term. Authors have not conducted the deep study. They just study the growth parameters and soil temperature and moisture content. I have some suggestions for authors.

1-The authors should rethink and revised the title of the manuscript. Because in title, authors have used word degradable film mulching, in rapid click its means that authors have used any organic made film mulching instead they have used polyethylene. The polyethylene plastic is one of the types of synthetic polymers and is difficult to degrade in soil in short time period. As the current study is performed for one cropping season. I suggest rethinking about the used word “degradable film mulching” and changing with more suitable words throughout the manuscript.

2-Line 26, In general, applying moderate amounts of sprayable degradable film at a rate of 2500 kg ha-1 presents an effective agronomic strategy for managing residual film contamination while maintaining stable cotton yield. These lines are the suggestions of the study, but it has negative effects for long term soil and environmental sustainability, what you suggest about next cropping season, should we continue this application for multiple years or just one time is enough?

3-Keywords: delete the words already mentioned in the title.

4- Introduction can be improved by providing the scientific justifications about the use of the sprayable degradable film such as concentrations and so on.

5-Line 77, correct the subsection heading.

6-Line 79 what 412 m a.s.l. is it MSL (Mean sea level).

7-Line 79 2014 in Xinjiang, China. Please add the total growing periods as for example Feb-July 2014.

8-Line 80 slope of 6‰. Double check the % symbol.

9-Line 97-98 SF1, 1900 kg of sprayable degradable film per ha; SF2, 2200 kg of sprayable degradable film per ha; SF3, 2500 kg of sprayable degradable film per ha; PF, plastic film; NF, no film mulching. Could you please justify, why you selected these plastic concentrations? What do mean by plastic film, does it mean plastic mulching?

10-Line 110 The amount of plastic film was the same as that used in general cotton fields. Please specify the amount.

11-Line 117 section 2.4, what were the soil properties before the start of the experiment, have you measured? It suggested adding in the manuscript.

12-Line 124 boll-opening stage, also mention the day of growth. How many plants were selected for root measurements per treatments? Please specify

13-Line 127 change the word killed with another one.

14-Line 169 revise the text format.

15-Table 1 revise the text format.

16-Revise the figures quality and text format.

Comments on the Quality of English Language

N/A

Round 2

Reviewer 2 Report

Comments and Suggestions for Authors

No comments

Author Response

Thank you very much for your efforts to improve the quality of our manuscripts!

Reviewer 4 Report

Comments and Suggestions for Authors

1- Line 59 to mitigate the soil pollution caused by plastic film residues, researchers have begun 59 adopting alternative mulching techniques to replace traditional plastic films (here add below reference).

You can add Agronomy. 2024; 14(3):548. https://doi.org/10.3390/agronomy14030548

2- Liine 88 is 3463.5 °C d. please confirm this.

3- Line 89- should Precipitation and annual air temperature were 212.8 mm and 22.3 °C, respectively during the cotton growing season (April-October) in 2014.

4-Line 93- The sprayable degradable film used in this study is developed. Delete the word degradable. Word degradable film mulching, in rapid click its means that authors have used any organic made film mulching instead they have used polyethylene. The polyethylene plastic is one of the types of synthetic polymers and is difficult to degrade in soil in short time period. REVISE THIS THORUGHOUT THE MANUSCRIPT.

5-Improve the figures quality.

6-There are grammar and text formatting issues; therefore, it is suggested to proof read the manuscript before resubmission.

Comments on the Quality of English Language

N/A
